# Inhibition of the Adenosinergic Pathway in Cancer Rejuvenates Innate and Adaptive Immunity

**DOI:** 10.3390/ijms20225698

**Published:** 2019-11-14

**Authors:** Juliana Hofstätter Azambuja, Nils Ludwig, Elizandra Braganhol, Theresa L. Whiteside

**Affiliations:** 1The Postgraduate Biosciences Program, Federal University of Health Sciences of Porto Alegre (UFCSPA), Porto Alegre 90050-170, RS Brazil; azambujaj@upmc.edu (J.H.A.); elizbraganhol@yahoo.com.br (E.B.); 2Department of Pathology, University of Pittsburgh School of Medicine, Pittsburgh, PA 15213, USA; ludwign@upmc.edu; 3UPMC Hillman Cancer, Pittsburgh, PA 15213, USA; 4Departments of Immunology and Otolaryngology, University of Pittsburgh School of Medicine, Pittsburgh, PA 15213, USA

**Keywords:** cancer, adenosine, CD73, immune system

## Abstract

The adenosine pathway plays a key role in modulating immune responses in physiological and pathological conditions. Physiologically, anti-inflammatory effects of adenosine balance pro-inflammatory adenosine 5’-triphosphate (ATP), protecting tissues from damage caused by activated immune cells. Pathologically, increased adenosine monophosphatase (AMPase) activity in tumors leads to increased adenosine production, generating a deeply immunosuppressed microenvironment and promoting cancer progression. Adenosine emerges as a promising target for cancer therapy. It mediates protumor activities by inducing tumor cell proliferation, angiogenesis, chemoresistance, and migration/invasion by tumor cells. It also inhibits the functions of immune cells, promoting the formation of a tumor-permissive immune microenvironment and favoriting tumor escape from the host immune system. Pharmacologic inhibitors, siRNA or antibodies specific for the components of the adenosine pathway, or antagonists of adenosine receptors have shown efficacy in pre-clinical studies in various in vitro and in vivo tumor models and are entering the clinical arena. Inhibition of the adenosine pathway alone or in combination with classic immunotherapies offers a potentially effective therapeutic strategy in cancer.

## 1. Introduction

The immune system plays an important role in the prevention as well as progression of malignancy [1]. On the one hand, it benefits the host by eliminating or neutralizing the tumor mass; on the other, it may be hijacked by the tumor to promote its progression. The recruitment of immune cells to the tumor suggests the presence of anti-tumor immunity. However, in the tumor microenvironment (TME), infiltrating immune cells become re-programmed by tumor-derived factors and assume a tumor-promoting phenotype [2]. 

The nucleoside adenosine (ADO) is involved in the regulation of diverse physiological and pathological processes [3,4,5,6,7,8,9]. ADO has recently emerged as a powerful immune checkpoint in the TME [10,11]. ADO acting directly on tumor cells promotes growth, survival, angiogenesis, chemoresistance, and metastasis [12,13]. However, under physiological conditions, ADO acting as an immuno-regulatory molecule protects normal tissues from inflammatory damage while in pathological conditions, it can impair anti-tumor immunity [14]. ADO attenuates functions of protective immune cells, including T cells and natural killer (NK) cells. It enhances the suppressive functions of regulatory T cells (Tregs), macrophages, and myeloid-derived suppressor cells (MDSCs), inducing cancer progression [15,16,17,18]. Here, we review approaches to blocking the adenosine pathway in cancer, removing the immune function brake and restoring the host’s ability to control tumor progression.

## 2. The ADO Pathway

Adenosine 5’-triphosphate (ATP) is a triphosphate nucleotide, and its main function is to provide energy to cells. For decades, this was the sole known ATP function. The history of the purinergic system began in 1929, when Drury and Szent-Gyorgi described the dilation of coronary blood vessels and hypotensive actions of purines in the heart and blood vessels [19]. Their findings contributed to establishing the current view of nucleotides and nucleosides as signaling molecules of the purinergic system. As such, they are involved in the regulation of various pathophysiological processes in the extracellular environments [20].

Purinergic signaling is characterized by the activity of extracellular purines (ATP, adenosine diphosphate (ADP), and ADO) or pyrimidines (uridine-5’-triphosphate (UTP) and uridine diphosphate (UDP)), which act as signaling molecules. Purinergic signaling serves as a communication system between cells. It is involved in a variety of mechanisms, including immune responses, inflammation, pain, platelet aggregation, proliferation, cell death, development, and neurotransmission [3,5,6,7,8,9,14,21,22]. Purines or pyrimidines exert their effects through interactions with specific membrane receptors called purinergic receptors or purinoceptors [5]. They comprise a group of enzymes that are involved in nucleotide degradation or nucleoside formation and, therefore, play a key role in the regulation of purinergic signaling [23].

### 2.1. Enzymes

The responses mediated by nucleotides and nucleosides upon binding to their respective receptor are catalyzed by enzymes called ectonucleotidases. These enzymes are responsible for the levels of nucleotides and nucleosides present in the extracellular microenvironment. Ectonucleotidases control the binding of nucleotides and nucleosides to their cellular receptors. As shows in Figure 1, this family of enzymes includes the ecto-nucleoside triphosphate-diphosphohydrolases (E-NTPDases), which catalyze the sequential degradation of ATP to ADP and adenosine monophosphate (AMP); the ecto-pyrophosphate-phosphodiesterases (E-NPP), which catalyze hydrolysis of ADP to AMP and of AMP to ADO; alkaline phosphatases (ALPs), which catalyze the degradation of ATP to ADP, ADP to AMP, and AMP to ADO; and, finally, the ecto-5’-nucleotidase (CD73), a 70-kDa glycosyl-phosphatidyl inositol-anchored protein on the plasma membrane. CD73 can be cleaved from the cell surface and act as a soluble enzyme, which irreversibly catalyzes the hydrolysis of AMP to ADO. In addition, adenosine deaminase (ADA) degrades ADO to inosine. The bioavailability of ADO is also regulated by nucleoside equilibrative transporters (ENTs) or concentrative nucleoside transporters (CNTs), which reside in the cell membrane and transport ADO into the cells. In the intracellular microenvironment, ADO is phosphorylated by ADO kinase (AdoK) and adenylate kinases into ADP [23].

However, ADO can also be generated by the non-canonical adenosinergic pathway by nicotinamide adenine dinucleotide (NAD+)-glycohydrolase/CD38 (NAD+ to ADP-ribose (ADPR)) and CD203a (PC-1) (ADPR to AMP) that is subsequently metabolized by CD73 to ADO. Therefore, CD73 represents the common link between the two adenosinergic pathways [24].

### 2.2. Receptors

The release of nucleotides and/or nucleosides into the extracellular microenvironment is accompanied by interactions with their respective receptors on cells, allowing for the purinergic signaling cascade to continue. Purinergic receptors are divided into two major groups, the P1R, which has the main endogenous agonist, ADO, and the P2R, which are sensitive to di- and triphosphate nucleosides, such as ATP, ADP, UTP, and UDP.

P1R, also called ADO receptors, are divided into four subtypes, A_1_, A_2A_, A_2B_, and A_3_, and all are coupled to G protein but differ in their affinity for ADO [25]. High-affinity receptors, A_1_R, A_2A_R, and A_3_R, bind ADO in the nanomolar range, whereas A_2B_R binds ADO in the micromolar range. Thus, at physiological concentrations of ADO, signaling is primarily mediated via A_2A_R, A_1_R, and A_3_R. A_2B_R are only activated when elevated levels of ADO are generated, such as in the inflammatory TME [26]. The A_1_R and A_3_R are coupled to the Gi or Go proteins, and their activation leads to a *decrease* in intracellular cyclic AMP (cAMP) levels, whereas A_2A_R and A_2B_R are coupled to Gs protein, resulting in *increased* levels of intracellular cAMP [25,27,28]. P1R are widely distributed among various cell types. They are expressed in the heart, lung, liver, testis, muscle, spinal cord, spleen, intestine, and brain [5]. In the immune system, these receptors are present in most cells and mediate the immunosuppressive and anti-inflammatory effects of ADO [18].

P2Rs comprise two categories of receptors, P2X and P2Y. P2YR are coupled to G protein and are metabotropic. P2XR are ionotropic and are divided into seven subtypes (P2X 1–7) that respond to ATP, whereas P2YR are subdivided into eight subtypes (P2Y 1, 2, 4, 6, 11–14) and are activated by ATP, ADP, UTP, and UDP, and are also sensitive to sugar nucleotides, such as UDP-glucose and UDP-galactose [29]. P2XR are broadly distributed in various cells, such as platelets, neurons, and muscle cells [30]. P2YR are found in a wide variety of organs and tissues: airway epithelium, different regions of the kidney, pancreas, adrenal gland, heart, vascular endothelium, skin, muscle, and various components of the nervous system, such as the cortex, hippocampus, and cerebellum [5].

## 3. ADO in Cancer

The role of ADO as a promoter of tumor progression is dependent on the activity and expression of CD73 in tumor cells. CD73 expression is elevated in different tumor types, including breast cancer [31], glioblastoma [32], F colorectal cancer [33], ovarian cancer [34], melanoma [34], gastric cancer [35], and bladder cancer [36]. Elevated CD73 expression levels significantly correlate with shorter overall survival in breast, ovarian, lung, and gastric cancer [37], and have been linked to cancer progression, migration, invasion, metastasis, chemoresistance, and neovascularization processes [13,38,39]. More importantly, ADO is now considered to be one of the most relevant immunosuppressive regulatory molecules in the TME [15,40,41]. Due to the favorable results seen in tumor models, targeting CD73 or ADORs has become a promising therapeutic approach in different types of human cancer. CD73 expression and ADO production by tumor cells have also been associated with the tumor progression, chemoresistance, migration, and angiogenesis, and these functions are summarized in Table 1, Table 2 and Table 3. 

## 4. ADO in the Immune System

It has been reported that ATP, ADP, and ADO play a key role in modulating immune responses [14]. In normal conditions, ATP is found mostly in the cytoplasm at the concentration of 3 to 10 mM, whereas in the extracellular compartment, ATP levels are low, ranging from 1 to 10nM. Extracellular concentrations of ATP, as well as those of other nucleotides, may increase in response to different stimuli or conditions, such as cell lysis, hypoxia, or inflammation [30]. High concentrations of ATP in extracellular fluids can be interpreted as an indicator of tissue damage, which can trigger an inflammatory response characterized by the secretion of pro-inflammatory cytokines [81]. On the other hand, ADO, which is released by tumor cells or formed by hydrolysis of ATP, generally acts contrary to extracellular ATP [82]. ADO concentrations in homeostatic situations range from 10 to 200nM, whereas in stress situations, ADO levels may be as high as 10 to 100μM [13]. Increased extracellular ADO concentrations occur in situations of ischemia, hypoxia, epithelial-to-mesenchymal transition, cytotoxic stress, or trauma [83]. ADO mediates immunosuppressive responses for the protection of tissues adjacent to the excessive inflammation against attacks by defense cells [84]. At such sites, immune cells express P1 receptors and ectonucleotidases [14]. These opposite functions of ATP/ADO in the control of the immune response are illustrated in Figure 2.

In human peripheral blood, CD73 is expressed on approximately 90% of monocytes, 50% of CD8+ T cells, 10% of CD4+ T cells, 75% of Tregs, and 2% to 5% of NK cells. While in mice, CD73 is expressed on approximately 90% of monocytes, 30% of CD8+ T cells, 10% of CD4+ T cells, 1% to 5% of Tregs, and 50% of NK cells [11]. Thus, the distinct CD73 expression profiles between human and murine immune cells may lead to distinct cellular responses. Therefore, data obtained from preclinical models should be carefully considered before making plans for the therapeutic use of CD73 inhibitors.

### 4.1. ADO in Macrophages

The mononuclear phagocytic system is among the major targets of ADO, and phagocytic cells are highly susceptible to ADO effects [89]. Macrophages express all subtypes of ADO receptors, although A_2A_R and A_2B_R are expressed at higher levels compared to A_1_R and A_3_R [88].

Macrophages are classified as M1, which are classically activated, and M2, which mediate anti-inflammatory functions [89,90,91], as shown in Figure 3A.

P1R activation has been shown to suppress M1 activation and proinflammatory cytokines release [88,92,94,95,96]. In contrast, its activation induces M2 activation [85,92,98] with IL-10 [96] and vascular endothelial growth factor (VEGF) production [97]. ADO effects on macrophages are summarized in Figure 3B.

Notably, blocking CD73 activity with a specific CD73 inhibitor (adenosine 5’-α, β-methylene-diphosphate) enhanced the M1 phenotype, diminished IL-4 and IL-10 production, and promoted pro-inflammatory cytokine release [87]. 

We demonstrated in M1 macrophages that ATPase activity was decreased, while M2-type macrophages increased ATP/ADP/AMP hydrolysis through increased expression of CD39/CD73. This has led to rapid ADO accumulation without alteration in the purinergic receptor expression [99]. ADO generated by M2 macrophages is implicated in a decreased proliferation of CD4^+^ T lymphocytes [100]. We studied the purinergic signaling following co-incubation of primary mouse macrophages with a mouse glioblastoma cell line. We found that A_2A_R and P2X7R activation was necessary for IL-10, monocyte chemoattractant protein-1 (MCP-1), and IL-6 release by macrophages after interaction with glioblastoma. The related cytokines modulated conversion of macrophages to the M2-phenotype [101] and decreased the activity of ectonucleotidases [102]. In ovarian cancer, tumor cells use CD39/CD73 enzymes to control macrophage migration [100]. A_2A_R stimulation has direct myelosuppressive effects that indirectly contribute to suppression of T cells and NK cells in primary and metastatic melanoma microenvironments, indicating that the blocking of the A_2A_R has the potential to enhance immune killing of tumors [103,104].

Future studies should focus on the potential of blocking CD73/P1Rs to control the inflammatory microenvironment, macrophage phenotypes, and pro-tumor activities in cancer. Blocking ADO production may favor M1 (antitumor)-type polarization and may inhibit M2-type polarization. Only limited data exist on strategies that could control macrophage polarization and tumor progression.

### 4.2. ADO in Lymphocytes

#### 4.2.1. CD4^+^ T Cells and NK

T cells can be activated as a result of antigen presentation by antigen-presenting cells (APCs). As a result, CD4^+^ T cells differentiate into T helper 1 (Th_1_) cells or T helper 2 (Th_2_) cells.

Human T lymphocytes were shown to express A_1_R, A_2A_R, A_2B_R, and A_3_R at different levels. A_2A_R were predominantly expressed and further upregulated upon stimulation [105]. Extracellular ADO inhibited T cell activation by APCs and modified T cell differentiation, cytokine production, and proliferation (Figure 4).

ADO signaling via A_2A_R negatively regulated the production of type 1 cytokines and enhanced the production of IL-10 [107,108,109] by cAMP/protein kinase A and caused Signal transducer and activator of transcription 5 (STAT5) dephosphorylation, which resulted in reduced IL-2R signaling in T cells [112] and inhibition of the nuclear factor kappa B (NF-kB) pathway [113]. This suggested that adenosinergic signaling via A_2A_R antagonism may be a promising target for activating anti-cancer immune responses. In addition, A_2B_R controls lymphocyte migration to the lymph nodes in mice, which facilitates the encounter of naive T cells with antigen presenting cells (APC) [114]. In this context, under activating conditions, human lymphocytes up-regulated A_3_ receptors [115], probably exerting protective effects against an immune attack.

A_1_R and A_2_R have the potential to regulate NK cell activity [116]. Human naive NK cells do not express significant levels of CD73 [117], but stimulated NK cells increased the expressions of A_2A_R, P2_X7_R, CD38, CD39, ENPP1, CD73, PANX1, and ENT1, and decreased ADA expression, suggesting that ADO is generated by canonical and non-canonical pathways after cellular activation [118]. The ADO effects on NK are presented in Figure 4. Mechanisms involve adenylyl cyclase, increased production of cAMP, and activation of PKA [75]. However, ADO also enhanced effector functions of NK cells in combination with IFN-alpha via A_3_R [119]. In addition, NK cells acted as regulatory cells, decreasing CD4^+^ T cell proliferation through ADO production via the ectoenzyme network, with a pivotal role for CD38 [120]. 

ADO suppressed metabolism and protein synthesis in NK cells, inhibiting oxidative phosphorylation and glycolysis [117]. Tumors utilize this strategy via the production of ADO to inhibit cytotoxic activity to NK cells and escape from the host immune system [121]. One strategy to circumvent NK cell suppression by the adenosinergic pathway would be to target A_3_R and/or A_2A_R because data show that A_3_R agonists and A_2A_R antagonists activate NK cells and further improve their anti-tumor effects in a melanoma mouse model in vivo [122,123].

#### 4.2.2. T Regulatory Cells

Tregs (CD4^+^ Foxp3^+^ regulatory T cells) play a key role in maintaining the control immune responses in human health and disease. Treg co-express CD39/CD73 on the surface and generate extracellular ADO, contributing to immunosuppressive activities [124]. Recently, it has been shown that lymphocytes can also generate ADO in a non-canonical way through ectoenzymes CD38 (an NAD^+^ nucleosidase), NPP1, and CD73 [125]. 

The ADO effects on Treg are summarized in Figure 4. ADO mediated the suppression of Tregs by PGE_2_ receptors expressed on T cells, leading to the upregulation of adenylate cyclase and cAMP activities [126]. Tumors use the adenosinergic pathway by increasing ADO production to promote Treg activity, aiming at an immunosuppressive microenvironment to escape immune surveillance and promote cancer growth [127]. 

#### 4.2.3. CD8^+^ T Cells 

Tumors utilize ADO to suppress CD8^+^ T cell functions and to avoid tumor rejection. Otha et al. reported that in A2AR−/− tumor-bearing mice, tumor-infiltrating CD8^+^ T cells mediated tumor rejection [128]. Consistent with this, stimulation of cytotoxic T lymphocytes with a P1R agonist decreased cell proliferation, IFN-γ production, and cytotoxicity [110,111]. After A_2A_R stimulation, CD8^+^ T cells produced higher levels of cAMP and decreased IFN-γ and IL-2 expression, decreasing maturation of CD8^+^ T cells [110]. These data suggest that A_2A_R-antagonists should be considered in therapeutic protocols, since this approach may improve antitumor immunity and control tumor growth. 

## 5. ADO in Exosomes

Exosomes are emerging as critical but poorly understood components of a complex communication network between the tumor and host cells. Exosomes are virus-sized extracellular vesicles deriving from the endocytic compartment of parent cells. They are secreted by normal and malignant cells and are present in all body fluids. Such molecules can serve as biomarkers of tumor progression and the immune response, or as predictors of response to therapies [129,130]. 

Exosomes derived from cancer cells were shown to carry CD39 and CD73 on their surface and exhibit potent ATP-AMP phosphohydrolytic activities. Multiple myeloma cell-derived exosomes release exosomes that are equipped with CD39/CD73 and with the enzymes that generate ADO via the non-canonical pathway (NAD+ /CD38/CD203a(PC-1)/CD73), and thus are able to generate ADO utilizing both the canonical and non-canonical pathways [131]. ADO produced by exosomes was shown to inhibit T- cell activation through A_2A_ R [132]. The exosome-mediated increase of ADO production is not only driven by direct production of ADO by exosomes but also indirectly by inducing/upregulating the expression of ADO pathway components on recipient cells. Prostate cancer-derived exosomes induced CD73 expression on dendritic cells, which led to an inhibition of tumor necrosis factor-alpha (TNFα) and IL-12 production by T lymphocytes in an ATP-dependent manner [133]. In addition, exosomes released from head and neck squamous cell carcinoma cells expressed CD39 and CD73 and increased the ADO production in Treg cells [134]. Treg-derived exosomes were also shown to carry biologically active CD73, and their production of ADO was considered to play a regulatory role for this cell type, stimulating immunosuppressive functions [135]. Thus, tumor-derived exosomes are emerging as a new mechanism of cancer-driver immune suppression that involves the ADO pathway.

## 6. ADO Pathway in Cancer Therapy

### 6.1. Targeting ADO Receptors in Cancer Cells

#### 6.1.1. A_1_R

The potential of targeting A_1_R for cancer treatment was mainly explored in breast cancer. In this context, estradiol, which has a critical role in breast cancer growth, increased A_1_R expression in breast cancer cell lines. Transfection with anti-A_1_R siRNA decreased cell proliferation, indicating that targeting the A_1_R receptor is beneficial in hormone-dependent breast cancer [136]. Furthermore, antagonism of A_1_R induced apoptosis of breast cancer cells by the upregulation of p53, caspase 3, 8, and 9 expression [50]. A_1_R antagonism also suppressed renal cell carcinoma growth/migration, induced apoptosis and cell cycle arrest in vitro, and reduced tumor volume in a nude mice subcutaneous model [137]. In addition, our data indicate that A_1_R is responsible for ADO-induced stimulation of gliomas proliferation in vitro [44]. However, the potential of A_1_R blockade still needs to be further explored, mainly in vivo in immunocompetent animal models, to prove its potential for future therapy in humans.

#### 6.1.2. A_2A_R

Antagonism of A_2A_R is currently the most widely used pre-clinical approach to cancer therapy. The A_2A_R blockade impaired lung adenocarcinoma tumor cells’ growth in vitro and inhibited human tumor xenograft growth in mice [57]. The A_2A_R blockade also protected against tumor metastasis and enhanced NK cell functions in an in vivo melanoma model [69]. Inhibition of serine/threonine-protein kinase B-Raf (BRAF) and mitogen-activated protein kinase (MEK) in combination with A_2A_R provided a significant reduction of tumor progression and metastasis formation in a melanoma mouse model [58]. The A_2A_R-deficient mice showed an increased intratumoral CD8+ T cell frequency and number [138] and had increased frequencies of tumor-associated NK cells in melanoma models [122]. Pharmacological targeting of A_2A_R increased CAR T cell efficacy [139]. Pharmacological blockade of A_2A_R by selective antagonists decreased tumor growth in a head and neck squamous cell carcinoma mouse model by reducing the population of Tregs inside the tumor, spleen, blood, and lymph node, and enhanced the anti-tumor response of CD8^+^ T cells by increasing the release of IFN-γ [56]. Dual blockade of programmed cell death protein 1 (PD-1) and A_2A_R significantly enhanced the expression of IFN-γ and granzyme B by tumor-infiltrating CD8^+^ T cells and, accordingly, increased growth inhibition and the survival of mice in a breast cancer model [140] and increased the efficacy of a dendritic cell-based cancer vaccine by increasing the production of IFN-γ and reducing IL-10 [141]. This suggests the strong association between antitumor effects of the A_2A_R antagonist and the immune system, and the potential of its blockade for the restoration of effective anti-tumor immune responses.

#### 6.1.3. A_2B_R

Selective antagonism of A_2B_R was shown to inhibit cell growth in prostate cancer cell lines [53]. Antagonizing A_2B_R increased colorectal cancer cell death in vitro via enhanced mitochondrial oxidative phosphorylation and reactive oxygen species (ROS) production [142]. A_2B_R antagonism in breast cancer also induced cell cycle arrest and apoptosis through the cyclin-, Bax/Bcl-2, and extracellular receptor kinase (ERK1/2) pathways [63]. In vivo, A_2B_R blockade also reversed immune suppression in the tumor microenvironment by reducing levels of IL-10, MCP-1, and MDSCs in melanoma lesions and increasing the frequency of CD8^+^ T lymphocytes and NK cells, and increasing the levels of Th_1_-like cytokines [42]. Enhancing anti-tumor immunity by inhibiting differentiation to Treg, which resulted in a suppression of metastasis in a melanoma mouse model in vivo, is another reported result of A_2B_R antagonism [59]. Cekic et al. reported that targeting A_2B_R the reduced growth of bladder and breast tumors in syngeneic mice, reduced metastases of breast cancer cells, increased tumor levels of IFN-γ, and enhanced dendritic cell activation, leading to an improved anti-tumor response [143].

#### 6.1.4. A_3_R

As mentioned above, A_3_R is involved in control of tumor progression. The potential of A_3_R activation for cancer treatment has been explored. A_3_R mediated apoptosis in human bladder cancer cell lines via protein kinase C in vitro [144]. It also decreased cell growth and cell motility in breast cancer cell lines [70,145]. In addition, agonism of A_3_R enhanced TNF-related apoptosis-inducing ligand-mediated apoptosis (TRAIL) in thyroid carcinoma cells via NF-kB activation [146], induced cell death in glioblastoma by increasing Ca^2+^ and ROS, and downregulated ERK and AKT [61]. In vivo, agonism of the A_3_R reduced tumor growth and breast tumor-derived bone metastasis in a rat preclinical model [66], colon carcinoma [147]. 

### 6.2. CD73

Considering the tumor-promoting effects of ADO in the TME, the enzymatic activity of CD73 appears to be a promising target for the treatment of cancer. Three approaches of CD73 inhibition have been described: Neutralizing antibodies, siRNA, or pharmacologic inhibition.

Recently, we demonstrated the potential for CD73 blockade using either specific siRNA sequences or a pharmacological inhibitor in controlling glioblastoma progression in vitro and in an orthotopic immunocompetent in vivo model using Wistar rats. We showed that inhibition of CD73 with both technologies decreases the amount of ADO in the cerebrospinal fluid and correlates with decreased tumor volume and increased lymphocytic infiltrate [44]. Similar findings were reported in a breast cancer immunocompetent mice model [148]. In breast cancer, anti-CD73 monoclonal antibody (mAb) inhibited metastasis formation [149]. This strategy may also be combined with standard chemotherapy, since in vivo evidence in a mouse ovarian carcinoma model indicates that combining docetaxel with anti-CD73 antibody therapy is more effective than using both treatments individually [48]. Studies demonstrated synergism between anti-CD73 and anti-PD-1 or anti-cytotoxic T-lymphocyte-associated protein 4 (CTLA-4) antibodies in pre-clinical models. A selective inhibitor of CD73 enhanced anti-CTLA4 mAb efficiency by improving the anti-tumor immune response, and reduced melanoma growth in vivo [150]. However, in order to translate these technologies from the bench to the bedside, it is necessary to think about how to deliver these drugs to the tumor and avoid adverse effects. In this context, the strategy that is under consideration involves the application of nanotechnology. Zhi et al. showed that intravenous administration of CD73-siRNA-loaded nanoparticles led to reduced tumor growth and metastasis formation, with improved mice survival in a breast cancer model [151]. Jadidi-Niaragh et al. showed that CD73-siRNA encapsulated into chitosan-lactate nanoparticles suppressed CD73 expression in breast cancer cell lines and protected the oligonucleotides against serum and heparin degradation, suggesting that this system may increase siRNA performance when administered to patients by increasing the half-life of the sequences [152]. We demonstrated the potential of a cationic nanoemulsion delivering an siRNA anti-CD73 via the nasal for glioblastoma treatment in vitro and in vivo [79]. Nevertheless, the strategy of blocking CD73 requires a carefully strategy, because despite its prominent role, CD73 is not the only enzyme capable of producing ADO in the TME. Therefore, a strategy combining the blockade of CD73 enzymatic activity in combination with ADO antagonists may be of interest and may potentiate the antitumor effect of a therapy targeting the adenosinergic signaling pathway. In recent years, many small molecule drugs or antibodies targeting the adenosinergic pathway have undergone or are undergoing clinical trials. Their targets include CD73, A_2A_R, and A_2B_R. These findings are summarized in Table 4. 

### 6.3. CD38

As cited above, ADO can also be generated by the non-canonical pathway. This pathway starts with CD38. Therefore, inhibition of the adenosinergic ectoenzyme function by the targeting of CD38 may lead to lower ADO levels in the TME. The first class of CD38-targeting antibody, daratumumab, is currently approved as a single agent and in combination with standards of care for the treatment of multiple myeloma [153]. Their antitumor effect is related to immunoregulatory activities. Daratumumab reduced the frequency of regulatory B cells, Tregs, and MDSCs, and enhanced helper and cytotoxic T cells [154].

### 6.4. Combined Therapies 

#### 6.4.1. Radiotherapy

Radiotherapy (RT)-induced damage leads to complex interactions between immune cells and tumors. RT has a dual effect, as it may activate innate and adaptive immune responses or increase immunosuppression by killing immune and tumor cells, inducing the accumulation of immunoregulatory mediators and cell types. Various observations from preclinical and clinical studies suggest that targeting radiation-induced immune deviation may offer promising opportunities for improvements of response to radiotherapy. In this context, the CD73-ADO axis system is currently seen as a major target [155]. However, the question about the effects of combined radiotherapy and anti-CD73/ADO antagonism is still open and awaits further studies.

#### 6.4.2. Chemotherapy

As shown in Table 1, the CD73-ADO axis plays a role in cancer chemoresistance. It has been reported that the blockade of CD73 and ADOR enhances the effect of chemotherapy [43,45,48,49]. This indicates that inhibition of the adenosinergic pathway contributes to the enhancement of chemotherapy. It has also been elucidated that combination therapy with anti-CD73 antibodies enhances the response in vitro to vincristine [45], temozolomide [44], docetaxel [48], doxorubicin [49], TRAIL [46], and anti-human epidermal growth factor receptor 2 (HER2) antibody [47]. Moreover, a higher efficacy has been reported with A_2B_R antagonists [42]. The potential mechanisms associated with ADO-mediated chemoresistance are dependent on P1R activation and expression and the activity of multidrug resistance-associated protein (MRP1). However, additional in vivo studies are necessary to elucidate the real potential of the anti-CD73/ADOR antagonism to improve chemotherapy responses.

#### 6.4.3. Immunotherapies

Checkpoint receptor blockade can lead to restoration of anti-cancer responses. However, not all patients respond to treatment, highlighting the need for further research to better understand tumor evasion mechanisms and to identify other targets that can effectively remove ‘brakes’ imposed on the immune response by the tumor. As described here, ADO is known to act through ADOR to negatively regulate T cells, macrophages, and NK cells responsible for anti-tumor responses in the TME, and thus targeting this pathway may prove to be a significant immunotherapeutic strategy. First, PD-1 blockade upregulates the expression of A_2A_R, making these cells more susceptible to ADO-mediated immunosuppression [140]. In this context, preclinical studies demonstrated a synergism between anti-CD73 and anti-PD-1 mAbs [156] or CTL-4 mAbs [150]. Synergism has also been observed between anti-PD-1 and A_2A_R antagonists [140,157], suggesting that combining CD73 and ADO as a therapeutic target might overcome immune system suppression in cancer and improve immunotherapeutic responses. 

## 7. Conclusions

The CD73-ADO axis, among the currently investigated anti-tumor strategies, has gained much attention as a novel immune checkpoint for cancer therapy. Numerous preclinical studies showed that ADO mediates pro-tumor as well as immunosuppressive activities. The potential of CD73/ADOR inhibition for controlling tumor growth and metastasis formation and activation of the immune system increases interest in targeting the ADO pathway for cancer treatment. Several clinical trials involving antibodies/inhibitors targeting CD73 and antagonists/agonists targeting ADOR in cancer patients are currently ongoing. 

## Figures and Tables

**Figure 1 ijms-20-05698-f001:**
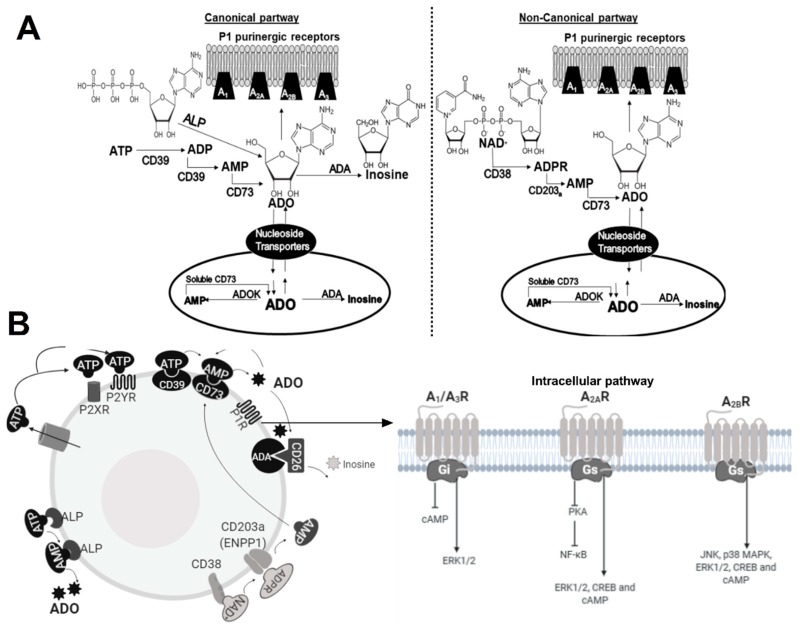
Extracellular and intracellular (canonical and non-canonical) adenosinergic pathways. In (**A**) biochemical interactions in the adenosine (ADO) pathway and in (**B**) cellular receptors in the ADO pathway.

**Figure 2 ijms-20-05698-f002:**
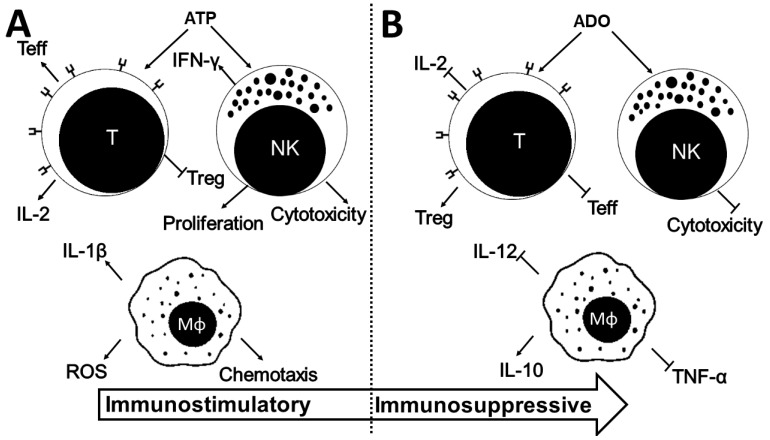
The opposite effects of ATP (pro-inflammatory) and ADO (anti-inflammatory) on immune cells (T cells, NK cells, and macrophages). In (**A**), ATP in extracellular fluids can be considered as a damage-associated molecular pattern (DAMP), which can trigger an inflammatory response characterized by proliferation, cytotoxicity, and the secretion of pro-inflammatory cytokines, such as interleukin-2 (IL-2), interferon-γ (INF-γ), and interleukin-β (IL-1β) [81]. In (**B**), ADO mediates immunosuppressive responses for the protection of tissues adjacent to inflammation from attacks by immune cells. In this case, ADO induces secretion of anti-inflammatory cytokines, such IL-10, and reduces secretion of pro-inflammatory cytokines, such as tumor necrosis factor-α (TNF-α), IL-12, and IL-2 [42,85,86,87,88].

**Figure 3 ijms-20-05698-f003:**
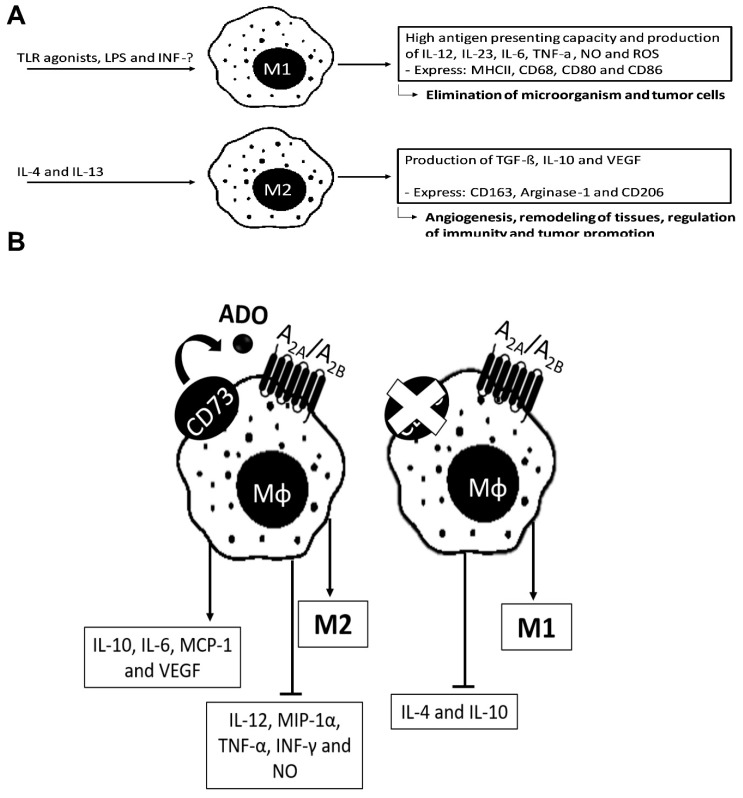
Effects of extracellular ADO on subsets of macrophages. In (**A**), differentiation of M1 and M2 and their characteristics is shown, including functions they mediate. In (**B**), the activation of the P1Rs on macrophages suppresses the M1 phenotype [88]. Activation of A_2A_R and A_2B_R inhibits production of interleukin-12 (IL-12) [92], interferon-γ (IFN-γ) [93], macrophage inflammatory protein-1 (MIP-1α) [94], tumor necrosis factor-α (TNF-α) [95], and nitric oxide (NO) [96] and induces IL-10 [96] and vascular endothelial growth factor (VEGF) production (*left side*). In contrast, suppression of CD73 activity enhances the M1 phenotype and blocks IL-4 and IL-10 production [87,97] (*right side*).

**Figure 4 ijms-20-05698-f004:**
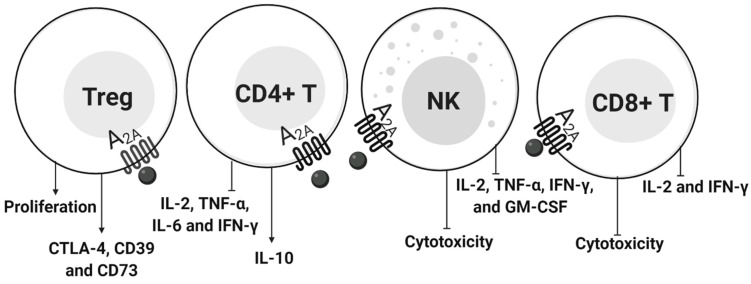
The cellular effects of extracellular ADO in the regulation of lymphocyte functions. After A_2A_R stimulation, regulatory T cells (Tregs) increased proliferation and expression of CTLA-4, CD39, and CD73, and inhibited CD8^+^ T cell proliferation [106]. CD4^+^ T cells decreased the production of IL-2, TNFα, IL-6, and IFNγ, and enhanced the production of IL-10 [107,108,109]. In NK cells, ADO suppresses the production of IL-2, TNF-α, IFN-γ, and granulocyte macrophage colony-stimulating factor (GM-CSF) and blocks their cytolytic activity [75]. In CD8^+^ T cells, ADO suppresses the production of IL-2 and IFN-γ and blocks their cytolytic activity [110,111].

**Table 1 ijms-20-05698-t001:** In vitro and in vivo studies of ADO chemoresistance activities reported in the literature.

Model	Main Result	Mechanism
**P1R antagonism**		
Melanoma in vivo	Inhibition of A_2B_R enhanced efficacy of dacarbazine	Reversed immune suppression in the TME [42]
Glioblastoma in vivo	Inhibition of A_2B_R enhanced efficacy of TMZ	A_2B_R [43]
**CD73 inhibition**		
Glioblastoma in vitro	CD73 KO increased efficacy of TMZ	ADO production [44]
Glioblastoma in vitro	CD73 KO reversed the MDR phenotype	A_3_R [45]
Leukemia in vitro	CD73 KO restored TRAIL sensitivity	Independent of CD73enzymatic activity [46]
Breast cancer in vivo	Anti-CD73 mab therapy enhanced efficacy of anti-ErbB2 mAb	Association of CD73 expression with TGF-β, EMT and HIF-1 [47]
Ovarian cancer in vitro and in vivo	Anti-CD73 mab therapy enhanced docetaxel response	Reverse the immunosuppression [48]
Breast cancer in vivo	CD73 inhibitor therapy enhanced efficacy of doxorubicin	Activation of immune response mediated by A_2A_R [49]

**Table 2 ijms-20-05698-t002:** In vitro and in vivo studies of pro and anti-tumor activities of ADO reported in the literature.

Model	Main Result	Mechanism
**P1R antagonism**		
Breast cancer in vitro	Inhibition of A_1_R induced apoptosis	Upregulation of p53 and caspases [50]
Colon carcinoma in vitro	Inhibition of A_2B_R suppressed tumor growth	A_2B_R [51]
Prostate cancer in vitro	Inhibition of A_2B_R suppressed tumor growth	A_2B_R [52,53]
Oral squamous cell carcinoma in vitro	Inhibition of A_2B_R suppressed tumor growth	A_2B_R [54]
Melanoma in vivo	Activation of P1R inhibited melanoma growth	Enhance immune killing of tumors [55]
HNSCC in vivo	Inhibition of A_2A_R suppressed tumor growth	Reduced Tregs population and enhanced the anti-tumor response of CD8+ T cells [56]
Lung adenocarcinoma in vivo	Inhibition of A_2A_R suppressed tumor growth	Prevented negative signaling in T cells and inhibited angiogenesis [57]
Melanoma in vivo	Inhibition of A_2A_R suppressed tumor growth	NK activation [58]
Melanoma in vivo	Inhibition of A_2B_R suppressed tumor growth	Reduced Tregs population and increased in CD4^+^ and CD8^+^ T cells [59]
**P1R agonism**		
Leukemia in vitro	Activation of A_3_R induced cell cycle arrest and apoptosis	Modulation of Wnt, β-catenin, GSK-β and AKT [60]
Bladder cancer in vitro	Activation of A_3_R induced cell cycle arrest and apoptosis	ERK and JNK activation [61]
Cancer cell lines	Activation of A_3_R induced cell cycle arrest and apoptosis	Downregulation of CDK4, cyclin D1 and upregulation of p53 [62]
Ovarian cancer in vitro	Activation of A_3_R reduced cell viability and induced cell cycle arrest	Downregulation of Cyclin D1 and CDK4 [63]
Renal cancer in vitro	Activation of A_3_R induced apoptosis	AMID upregulation [64]
Glioblastoma in vitro	Activation of A_3_R induced cell death	ERK and AKT downregulation [62]
Lung cancer in vitro	Activation of A_3_R induced cell death	Mediated by caspases upregulation [65]
Breast cancer in vivo	Activation of A_3_R inhibited tumor proliferation	Not reported [66]
**CD73 inhibition**		
Glioblastoma in vitro and in vivo	Knockdown of CD73 decreased glioma growth	Stimulation of AKT/NF-kB pathways [44]
**CD73 overexpression**		
Medulloblastoma in vitro and in vivo	Reduced proliferation and vascularization	Mediated by A_1_R [67]

**Table 3 ijms-20-05698-t003:** In vitro and in vivo studies of the ADO role in tumor migration, invasiveness, and angiogenesis as reported in the literature.

Model	Main Result	Mechanism
**P1R antagonism**		
Melanoma in vitro	Reduced angiogenesis	A_2B_R blockade impairs IL-8 production, whereas blocking A3R decreases VEGF [68]
Breast cancer and melanoma in vivo	A_2A_R blockade reduced metastasis	Enhanced NK cell maturation and cytotoxicity [69]
**P1R agonism**		
Breast cancer in vitro	Activation of A_3_R induced migration	Not reported [70]
Colon cancer in vitro	Enhanced migration	A_2B_R and A_3_R activation and regulation HIF-1alpha/VEGF/IL-8 via ERK1/2, p38, and AKT [71]
**CD73 inhibition**		
Ovarian Carcinoma in vitro	CD73 inhibitor blocked migration	Not reported [72]
Glioblastoma in vitro	CD73 KO decreased migration and invasion	Altered MMP-2 and Vimentin expression [44]
Breast cancer in vitro	Anti-CD73 mab therapy inhibited migration, invasion and adhesion	EGFR and IL-8 [73]
Breast cancer in vivo	Anti-CD73 mab therapy decreased lung metastases	Activation of NK cells, CD8+ T and IFNγ by A_2B_R [74,75]

Melanoma in vitro and in vivo	CD73 inhibitor decreased adherence of cells and enhanced migration and invasion	Via P1R [76]

Breast cancer in vitro and in vivo	Anti-CD73 mab therapy inhibited migration metastasis in vivo	CD73 expression promoted autophagy [77]

Hepatocellular cancer in vitro and in vivo	CD73 KO inhibited migration, invasion and metastasis	A_2A_R activates Rap1, P110β, and PIP3 production by AKT [78]
Glioblastoma in vivo	CD73 KO inhibited angiogenesis	Not reported [79]
**CD73 overexpression**		
Cervical cancer in vitro	Promoted migration; and high concentration inhibited migration.	Upregulation of EGFR, VEGF, and AKT [80]

**Table 4 ijms-20-05698-t004:** Clinical trials ^a^.

NCT Number	Phase	Year	Type of Cancer	Drug Name	Target
NCT00879775	Phase 2	2009	Cancer	Caffeine	P1R antagonist
NCT024031093	Phase 1/2	2015	Non-small Cell Lung Cancer (NSCLC)	PBF-509	A_2A_R antagonist
NCT02655822	Phase 1	2016	Advanced Cancers	CPI-444	A_2A_R antagonist
NCT03274479	Phase 1	2018	Locally Advanced or Metastatic NSCLC	PBF-1129	A_2B_R antagonist
NCT00790218	Phase 1/2	2009	Hepatocellular Carcinoma	CF102	A_3_R antagonist
NCT01987999	Phase 2	2013	Prostate Cancer	Acetogenins	ATP inhibitor
NCT02503774	Phase 1	2015	Solid Tumors	MEDI9447	CD73
NCT03267589	Phase 2	2017	Relapsed Ovarian Cancer	MEDI9447	CD73
NCT03616886	Phase 1/2	2018	Triple Negative Breast Cancer	MEDI9447	CD73
NCT03549000	Phase 1	2018	Advanced Malignancies	NZV930	CD73
NCT03381274	Phase 1/2	2018	NSCLC	MEDI9447	CD73
NCT03454451	Phase 1	2018	Cancer	CPI-006	CD73
NCT03835949	Phase 1	2019	Advanced or Metastatic Cancer	TJ004309	CD73
NCT03875573	Phase 2	2019	Luminal B Breast Cancer	oleclumab	CD73

^a^ A current list of clinical trials investigating the role of targeting purinergic signaling in cancer.

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
