# Peer review of "Inhibition of the Adenosinergic Pathway in Cancer Rejuvenates Innate and Adaptive Immunity"

_ijms, 2019, doi:10.3390/ijms20225698_

Round 1

Reviewer 1 Report

Authors reviewed effects of adenosinergic pathway on innate and adaptive immunity and pharmacological inhibitors for the pathway to enhance anti-tumor immunity.

I have some requests.

Drs. Ohta A and Sitkovsky MV reported that negative roles of A2A adenosine receptor in T-cell immune responses between 2006-2009 (PMID: 16916931, 18829471, 19076726, and 19843934). However, the authors referred only their 2012 paper as #125. Because those papers should be one of the primary studies reported about physiological functions of A2A receptor, the authors should mention them in the appropriate sections.

Also, they reported the effects of ADO in CD8+ T-cells and invariant NKT cells. If this review article also include the effects of ADO on the cell population, it would get much better.

Author Response

Answer to Reviewer 1

Comments:
1. “The authors should mention [manuscripts by Ohta A and Sitkovsky MV] in the appropriate sections.”

Response: Thank you for your suggestion. The references were added in the final version of the manuscript [125-128]. (page 10, lines 258-268).

“they reported effects of ADO in CD8+ T-cells and invariant NKT cells. If this review article also included the effects of ADO on CD8+ T cell population, it would get much better.”

Response: A new section about the ADO effects on CD8+ T-cells was added (Section 4.2.3. CD8+ T cells). Figure 4 was revised and now includes also the CD8+ population.

Overall

We would like to draw your attention to Figure 2.  The formatting has compressed the arrow at the bottom so that the writing (“immunostimulatory” and “immunosuppressive”) are not legible.  We encourage you to resize the figure so that this information is conveyed.

Reviewer 2 Report

The manuscript “Inhibition of the adenosinergic pathway in cancer rejuvenates innate and adaptive immunity” represents a very interesting compendium about the function of adenosinergic machinery in cancer and immune cells and highlights therapeutic attempt to inhibit it for activation of cancer immune response or for inhibition of immunosuppression induced by adenosine. This manuscript is nicely written and easy to follow. Besides it is very good illustrated and accompanied with useful tables. I guess this review will be no doubt of interest for researches in the field after a small improvement.

Specific comment:

May be the authors point clear the difference between the surface CD73 expression on human and murine immune cells. Since a low surface expression of CD73 might be an obstacle for immunotherapy using CD73 inhibition. The authors may discuss it shortly.

Author Response

Answer to Reviewer 2

“The manuscript “Inhibition of the adenosinergic pathway in cancer rejuvenates innate and adaptive immunity” represents a very interesting compendium about the function of adenosinergic machinery in cancer and immune cells and highlights therapeutic attempt to inhibit it for activation of cancer immune response or for inhibition of immunosuppression induced by adenosine. This manuscript is nicely written and easy to follow. Besides it is very good illustrated and accompanied with useful tables. I guess this review will be no doubt of interest for researchers in the field after a small improvement”.

 Response: Thank you.

May be the authors point clear the difference between the surface CD73 expression on human and murine immune cells. Since a low surface expression of CD73 might be an obstacle for immunotherapy using CD73 inhibition. The authors may discuss it shortly.

Response: The differences between CD73 expression on human and murine immune cells were added followed by a short discussion (page 6, lines 155-162).

Overall

We would like to draw your attention to Figure 2.  The formatting has compressed the arrow at the bottom so that the writing (“immunostimulatory” and “immunosuppressive”) are not legible.  We encourage you to resize the figure so that this information is conveyed.

Reviewer 3 Report

The manuscript by Hoffstätter Azambuja et al. reviews the biology of the adenosinergic pathway, the role of adenosine in shaping the immunological profile in the tumor microenvironment and highlights potential targets for cancer therapy within the adenosinergic pathway.

The first part of the review is an excellent and very comprehensive description of the adenosine pathway biology, cellular targets, with well-designed figures focusing especially on extracellular ADO generation, key enzymes and key receptors involved in the process. Though the intracellular signaling is less well explored. A figure might help here also.

The second part is less convincing. Potential therapeutic targets are described in detail but in a descriptive manner only. Less emphasis is placed on explaining the known/suspected mechanisms how ADO (or components of the ADO pathway) interferes with existing antitumor modalities. For example it is stated several times in the manuscript that ADO increases chemoresistance of tumors. However, the authors do not elaborate on the potential mechanisms how this is achieved. It is not specified the effect of which chemotherapeutic agents is affected by ADO (apart of cisplatin, which is briefly mentioned).

Radiotherapy is still a major therapeutic modality for cancer treatment. Thus, any combined treatment modality has to explore potential interactions with radiotherapy. Nowadays, in the light of emerging data on the effect of radiotherapy on the immune system it is a rather simplistic interpretation that radiotherapy is an immune suppressive agent. The authors should discuss in more detail the role of ADO and various components of the adenosinergic pathway in modulating tumor radiosensitivity, the relation between ADO and tumor hypoxia and how this impacts tumor immunogenicity.

Author Response

Answer to Reviewer 3

“While the extracellular signaling is well presented, the intracellular pathway is not. A figure might help here also”.

Response: The figure 1 was revised and now also shows the adenosine intracellular signaling.

“The authors do not elaborate on the potential mechanisms how this is achieved. It is not specified the effect of which chemotherapeutic agents is affected by ADO”.

Response: The insights about the mechanisms involved in ADO-mediated chemoresistance were added (page 13, lines 418-419).

“Nowadays, in the light of emerging data on the effect of radiotherapy on the immune system it is a rather simplistic interpretation that radiotherapy is an immune suppressive agent.”.

Response: A dual effects of radiotherapy on the activation or inhibition of the immune system were discussed (page 13, lines 402-405).

“The authors should discuss in more detail the role of ADO and various components of the adenosinergic pathway in modulating tumor radiosensitivity.”.

Response: Unfortunately, the effects of the adenosinergic pathway on radiosensitivity have not yet been determined and remain open. In this context, any discussion about this subject would be just speculation. Therefore, here, we highlight the need for further studies

Overall

We would like to draw your attention to Figure 2.  The formatting has compressed the arrow at the bottom so that the writing (“immunostimulatory” and “immunosuppressive”) are not legible.  We encourage you to resize the figure so that this information is conveyed.

Round 2

Reviewer 3 Report

Reviewer's comments were addressed in an acceptable way.